# Study on Shear Resistance Property of a New PBL Connector with Steel–Rubber Tenon

**DOI:** 10.3390/ma16062291

**Published:** 2023-03-13

**Authors:** Wenru Lu, Donghui Li, Yuanming Huang, Jun Wu

**Affiliations:** 1School of Civil Engineering, Henan University of Technology, Zhengzhou 450001, China; 2Henan Urban Planning Institute & Corporation, Zhengzhou 450044, China; 3Henan Key Laboratory of Grain and Oil Storage Facility & Safety HAUT, Zhengzhou 450001, China; 4Highway School, Chang’an University, Xi’an 710064, China

**Keywords:** combined steel–concrete structure, new PBL connector, steel–rubber tenon, shear resistance, finite element analysis, parametric discussion

## Abstract

In order to improve the shear resistance and structural ductility of the perfobond rib (PBL) connector, a new PBL connector with steel–rubber tenon is proposed in this study, which aims to increase the shear load capacity of the connector while improving the ductility of the connector. First, models of new PBLs are established based on the validated finite element method, and their mechanical properties are compared with other shear connectors. The results show that the stiffness and shear load capacity of the proposed connector can be effectively improved when the steel ring is added, where the shear stiffness can be reduced, and the deformation capacity of the specimen can be improved when the rubber ring is added. When a steel ring with a thickness of 5 mm and a rubber ring with a thickness of 5 mm are involved, the shear load capacity of the connector with steel–rubber tenon is increased by 13.7%, and the shear stiffness is reduced by 37.3% compared to the conventional concrete tenon connector, while the ductility is increased by 75.1% compared to the connector with steel ring tenon. Subsequently, as for the connectors with steel–rubber tenon, the effects of the thickness of the steel ring, the thickness of the rubber ring, the diameter of perforated rebar, the strength of concrete and the strength of perforated steel plate are analyzed based on the finite element model of a PBL. The results show that an increase in the thickness of the steel ring, the diameter of the perforated rebar and the strength of the concrete will cause an increase in the shear stiffness and shear load capacity of the connector; however, an increase in the thickness of the rubber ring can cause a decrease in the shear stiffness and shear load capacity of the connector, while a change in the strength of perforated steel plate has little effect on the shear stiffness and shear load-carrying capacity. Finally, based on the finite element parametric analysis results and the damage mechanism of the proposed connector, a calculation equation applicable to the PBL connector with steel–rubber tenon is presented to predict the shear load capacity of the connector.

## 1. Introduction

With the rapid development of steel–concrete composite structures in bridges and buildings, shear connectors between steel and concrete structures have also received more attention [1,2]. Among them, the most widely used shear connectors are mainly perfobond rib (PBL) connectors [3] and stud shear connectors [4]. Compared with stud shear connectors, the perfobond rib (PBL) connector has received much attention due to its superior mechanical properties and long service life [5,6,7]. As for bridge engineering [8,9], Yang et al. [10] investigated the mechanical properties of the PBL connector group; Zhang et al. [11] studied the fatigue damage performance of the PBL connector. As for building engineering [12,13], Jin et al. [14] presented a new PBL connector between the frame and the shear wall. Moreover, the finite element method (FEM) is important in simulating the PBL connector [15].

To further improve the shear resistance of the PBL connector and increase the ductility, a new PBL connector with steel ring–rubber tenon is proposed in this study. However, the theoretical guidance and application for the design calculations of the new PBL connector are absent. The results of the traditional PBL connector and some new PBL connectors can provide an important basis and reference for the proposed PBL connector. Professor Leonhardt developed perforated steel plate connectors in Germany in 1987 and successfully applied them to the Caroni Bridge in Venezuela [16]. For many years, researchers have conducted a lot of research on perforated steel plate connectors to explore the shear performance of perforated steel plate connectors under the influence of various parameters. Ahn et al. compared the static performance of PBL connectors with and without perforated rebar. The results of the study showed that the shear load capacity of a connector with perforated rebar was twice that of a connector without it. Subsequently, in 2010, through experimental study and parametric analysis, the formula for calculating the shear load capacity of the PBL connector was proposed, considering the influence of the height, spacing and arrangement of the perforated steel plate [17,18]. Since then, He et al. have studied the mechanical properties of PBL connectors through push-out tests and proposed a formula for calculating shear capacity [19]. Hu et al. analyzed the influence of various factors on the shear capacity of PBL connectors through push-out tests and, on this basis, proposed a formula for calculating the shear capacity of PBL connectors [20,21]. Through numerical simulation and parameter analysis, Shi et al. found that the strength of concrete, the diameter of the steel bar and the number of openings significantly affect the force of PBL connectors; however, the diameter of the openings has little effect on it [22]. Akinori et al. studied the influence of the connector’s internal components on the shear performance of the connector through push-out tests and numerical simulation [23]. Su et al. conducted a regression analysis of the results of the push-out test and discussed the influence of concrete and transverse reinforcement on the shear performance of the inner diaphragm of the perforated steel plate shear connector [24]. Based on the push-out test, Wang et al. analyzed the mechanical properties and important influencing factors of PBL connectors, proposed the corresponding shear capacity formula, and further deduced the load–slip curve expression of PBL connectors [25]. Several scholars, including Vianna, Li, Zheng, Lin and Wang et al., analyzed the damage mode of the connector as well as the load–slip relationship by controlling each influencing parameter’s variable based on push-out tests and numerical simulations [3,26,27,28,29], while studying the shear resistance of PBL connectors as well. Suzuki and Kim et al. considered the state of the shear connector subjected to repeated loading in the actual operation and studied it through extensive push-out tests and numerical simulations, and found that the PBL connector was superior to the conventional pinned connector under this loading condition, and proposed their design of residual shear strength discount factors and design equations for the shear strength under cyclic loading [30,31,32]. Through the study of PBL connectors by many scholars, it can be found that the calculation method of shear capacity is a key point for evaluating the shear performance of PBL connectors, and the calculation method of shear capacity can be deduced by the combination of numerical simulations and parameter analyses.

In recent years, as the PBL connector has been intensively studied, researchers have gradually started to improve the material’s mechanical properties and develop new types of connectors simultaneously. Duan and He et al. investigated the structural performance of the UHPC-steel combination PBL connector based on concrete materials and found that the addition of UHPC reduced the surface cracking of concrete and strengthened the shear-bearing capacity of the connector through push-out tests and theoretical analysis [33,34]. Peng et al. proposed an SFRCC-steel composite PBL connector based on steel fiber-reinforced concrete. Through the combination of push-out tests and numerical simulation, it was found that steel fiber can prevent the development of concrete cracks. At the same time, according to the test results, it was found that the pore size, thickness and yield strength of the steel plate have a great influence on the shear capacity of the PBL connector [35]. Zheng et al. proposed a long-hole-type PBL connector and a slotted-type PBL connector while considering the hole’s structure to improve the shear-bearing capacity of the connector and facilitate construction. Zheng et al. verified that the connector had good shear and pullout resistance based on push-out tests, numerical simulations and parametric analyses, respectively [36,37,38]. In particular, in order to ensure the ductility of shear connectors, the deformation and slip capacity of shear connectors were improved. Xu and Zhang et al. proposed a combination connector with a pinned rod section with natural rubber and demonstrated it by push-out tests, and it was found that this combination significantly improved the ductility of the connector and reduced the shear stiffness of the connector [39,40]. Subsequently, Liu et al. applied rubber rings to PBL connectors and found that this combination could effectively improve the ductility of the connector; however, its shear load capacity was reduced [41].

In order to propose a novel PBL connector with higher performance, a new PBL connector with steel–rubber tenon is proposed in this study, which aims to increase the shear load capacity of the PBL connector while reducing the shear stiffness. From the above literature review, it can be known that there is currently no literature available on the force performance of this new PBL connector. To fill this blank, in this paper, the PBL connector is modeled, its shear resistance is investigated, and the static shear resistance of the connector is studied based on clarifying the load-bearing mechanism of the new connector. According to its force characteristics, the thickness of the steel ring and rubber ring, the diameter of perforated rebar, and the strength of concrete and perforated steel plate are set as variables through a nonlinear finite element simulation in the parameter analysis, and according to the results of the parameter analysis, the shear-bearing capacity calculation method of the connector was built.

## 2. New PBL Connector with Steel–Rubber Tenon

The practical application of the new PBL connector with steel–rubber tenon is shown in Figure 1, which adds a steel ring to the outside of the perforated rebar to increase its cross-sectional shear area. This approach can save a lot of steel compared to the method of increasing the diameter of perforated rebar. A natural rubber ring is added to the outside of the steel ring to reduce the local shear concentration behavior and reduce the shear stiffness of the connector. Meanwhile, steel ring and rubber ring compositions contain the steel–rubber tenon, which can realize the combination of beams based on standard components of the specialization, scale, information production and construction, and therefore can adapt to the development characteristics of the industrialization of construction and the application of broad prospects.

## 3. Design of New PBL Connector Elements with Steel–Rubber Tenon

For the shear resistance of the new PBL connector, the push-out test and the beam test are both used to test it. The connection is used in steel–concrete composite beams, and if the test is carried out using the composite beam and its actual state of stress, it is consistent with the forces on the connector. However, this test method is expensive, the test process is complex, and it is difficult to carry out a parametric analysis. Therefore, scholars are currently using the launch test to examine the shear-bearing performance of the connector, which is a low-cost test method that can be used to obtain safe experimental data, which basically conforms to the state of the connector. At present, national norms also use the results of the launch test as the basis for assessing the shear-bearing capacity of the connector. Therefore, push-out tests are proposed to test the static shear capacity of the new connector. In addition, many push-out tests of PBL connectors have been carried out by domestic and international scholars. In this paper, the authors aim to test the force performance of the key construction of this new connector, i.e., the steel–rubber tenon for improving the test efficiency, while the embedded push-out test model is built. The dimensional configuration of the specimen is shown in Figure 2.

The test piece comprises HRB400 grade hot-rolled rebars (20 mm diameter perforated rebar, 16 mm diameter transverse and vertical rebars, and 16 mm erection rebars). The steel ring comprises Q235 steel with a 20 mm inner diameter, 5 mm thickness and 50 mm length. The rubber ring comprises natural rubber with a 30 mm inner diameter, 5 mm thickness and 50 mm length. The perforated steel plate comprises Q345 steel with a 20 mm thickness, and the concrete slab is made of C60 concrete with 600 × 600 × 450 mm dimensions. The rubber ring comprises natural rubber with an inner diameter of 30 mm, thickness of 5 mm and length of 50 mm, and the perforated steel plate comprises Q345 steel with a thickness of 20 mm, while the concrete slab is made of C60 concrete with a size of 600 × 600 × 450 mm and is equipped with a layer of reinforcement mesh whose main purpose is to restrain the concrete from forming a reinforced concrete structure.

## 4. Finite Element Analysis

### 4.1. Cell Type Selection and Meshing

In order to investigate the force and shear load capacity of the new PBL connector with steel–rubber tenon, a refined simulation was carried out by the nonlinear finite element analysis software ABAQUS/CAE 2021, as shown in Figure 3.

It can be seen in Figure 3 that the model contains seven parts, which are the rigid base, concrete, perforated steel plate, perforated rebar, distribution reinforcement, steel ring and rubber ring, respectively. The concrete, perforated steel plate, perforated rebar, steel ring and rubber ring are divided using C3D8R units, which are less prone to shear self-locking and have less influence on the accuracy of the results when the mesh is distorted and deformed, in addition to the displacement results being more accurate as well. The base of the platform is simulated using R3D4 rigid units, considering enough stiffness. The main purpose of the distribution reinforcement is to restrain the concrete from forming a reinforced concrete structure. Thus, T3D2 cells are used in the division of the units, and only their effects on tension and compression are considered. The main shear zones, such as the concrete tenon and perforated steel plate near the opening, have meshed more intensively, describing their mechanical behavior more accurately.

### 4.2. Boundary Conditions and Loading Methods

As shown in Figure 3, a reference point was taken at the center of the rigid base and given a rigid body constraint, and the translational and rotational constraints were applied to this reference point in all directions. The perforated rebar and concrete, the perforated rebar and steel ring, and the steel ring and rubber ring are all tied together using the tie constraints without regard to mutual slippage. The distribution reinforcement is built into the concrete using the embedded region restraint and deformed the nodes together with the concrete, ignoring the slip. Contact pairs were established between the perforated steel plate and the concrete and between the concrete and the base and were simulated using surface-to-surface contact. No friction was considered between the perforated steel plate and the concrete, and further, to avoid penetration of the primary and secondary surfaces, the normal behavior was simulated using hard contact, which allows the contact areas to separate. The tangential behavior between the concrete and the base was simulated using a penalty function with a friction coefficient of 0.3 [41]. Finally, the model is displacement–loaded: a reference point was established above the geometric center of the upper surface of the steel beam, and all the degrees of freedom on the upper surface of the beam were coupled to this reference point, which is then displacement–loaded.

### 4.3. Material Modeling

#### 4.3.1. Constitution of Concrete

The concrete principal structure relationship was adopted from the concrete damage plasticity model, while the stress–strain relationship for uniaxial compression was adopted from the recommended values in CEB-FIP MC 2010 [42], and the nonlinear principal structure relationship, provided by CEB-FIP MC 1990, was considered for the concrete tensile behavior [43]. The concrete compressive stress–strain curve and tensile stress-crack width relationship curves are shown in Figure 4.

#### 4.3.2. Constitution of Steel

Both steel plates and rebars are modeled using an ideal elastic–plastic trifold model, considering hardening [37], as shown in Figure 5. The material properties are presented in Table 1.

#### 4.3.3. Constitution of Rubber

For natural rubber, as an isotropic hyperplastic material, strain potential energy is used to simulate its intrinsic structural relationship, according to the uniaxial tensile test of rubber proposed in [41], combined with the strain potential energy model provided by ABAQUS, considering the rubber ring in the model is wrapped by concrete and steel ring. This is a simplified polynomial chosen to simulate the rubber intrinsic structural relationship. The rubber intrinsic structure parameters are summarized in Table 2.

### 4.4. Validation of Finite Element Simulation Results

In order to verify the correctness of the finite element simulation method in this section, the PBL connector in [41] was simulated and verified first.

The simulation results show that the damaged form of the connector in the finite element is consistent with the test, and the deformation of the perforated steel plate and rebar when the specimen reaches the shear load capacity is illustrated in Figure 6.

The load–slip curve obtained from the simulation is a better fit than the test, as shown in Figure 7. The calculated yield load (*V_y_*) is 477.8 kN, and the shear capacity (*V_u_*) is 591.9 kN, which is only 2.8% and 3.4% different from the experimental results (*V_y_* = 491.7 kN and *V_u_* = 555.2 kN), respectively, verifying the high accuracy of the finite element simulation method in this section.

## 5. Shear Mechanism Analysis

Based on the finite element analysis method introduced above, models of the PBL connector contain a steel–rubber tenon; the PBL connector contains a steel ring tenon, and the conventional PBL connector is developed, as shown in Figure 8. The load–slip curves for the three connectors under the ultimate load are simulated as presented in Figure 9, where the yield slip (*S_y_*) is defined as the relative slip corresponding to the yield load (*V_y_*), and the shear stiffness (*K_s_*) is the ratio of the yield load (*V_y_*) to the yield slip (*S_y_*).

As can be seen from Figure 9, the load–slip curve is divided into three stages. In the first stage, the slope of the curve is large, which shows the linear elasticity and high stiffness behavior without a large slip. In the second stage, the curve trend is arc-shaped, which shows a nonlinear and gradual decrease in stiffness behavior, where the load increases slowly with the slip, and at the point of the extreme value of the curve (yield load *V_y_*), the load starts to show a slow. In the third stage, the load slowly increases with the slip until the specimen reaches damage.

For the traditional PBL connector, the load is applied to the perforated steel plate at the beginning of loading, and the shear force is transmitted through the perforated steel plate to the concrete tenon and the perforated rebar. With the concrete tenon, the concrete at the ends of the concrete tenon and the perforated rebar can withstand the shear force. As the load increases and the concrete tenon is damaged, the shear force on the concrete tenon decreases, and the shear force is mainly carried by the perforated rebar.

Figure 9 shows that yield load (*V_y_*) of the conventional PBL connector is 458.6 kN, the shear load capacity (*V_u_*) is 539.8 kN, and the shear stiffness (*K_s_*) is 158.2 kN/mm. The steel ring in the PBL connector with the steel ring tenon can increase the cross-sectional shear area through the steel while only increasing a small amount of steel. The shear stiffness (*K_s_*) of the PBL connector with steel ring tenon is increased by 9.8%, and the shear load capacity (*V_u_*) is increased by 9.2% compared to the concrete tenon member. The shear stiffness of the PBL connector with steel–rubber tenon is reduced by 37.3% compared to the concrete tenon member (*K_s_*) at the beginning of loading due to the small stiffness of the natural rubber, while the shear load capacity (*V_u_*) is increased by 13.6% due to the presence of the steel ring tenon. The yield load (*V_y_*) for the PBL connector with steel ring tenon and the steel–rubber tenon are 103.4% and 108.4% of the conventional PBL connector, respectively, and the shear load capacities are 109.2% and 113.7% of the conventional PBL connector, respectively.

Therefore, it can be concluded that the inclusion of the steel ring can effectively improve the shear resistance of the specimens. In addition, the yield slip (*S_y_*) of the PBL connector with steel–rubber tenon is 175.1% of the PBL connector with steel ring tenon, which indicates that the setting of the rubber ring can significantly improve the ductility of the specimen. At the same time, the yield slip (*S_y_*) of the PBL connector with steel–rubber tenon is 164.9% of the conventional PBL connector, which indicates that the steel–rubber tenon set-up can still significantly improve the ductility of the specimen.

## 6. Parametric Study

### 6.1. Proposal Selection

A parametric study was carried out on the proposed PBL connector with steel–rubber tenon, as described in this section. Twenty-seven finite element models were designed with different influencing parameters, such as *t_s_* being the thickness of the steel ring, *t_r_* being the thickness of the rubber ring, *d_r_* being the diameter of perforated rebar, *ƒ_cu_* and *ƒ_sy_* are the strength of concrete and steel plate, respectively, and the results of the parametric study are summarized in Table 3.

### 6.2. Effect of Steel Ring Thickness

The load–slip curves for the PBL connector with steel–rubber tenon at different steel ring thicknesses and the stress variation in the various components of the specimen are illustrated in Figure 10.

From Figure 10a, it can be seen that by keeping the diameter of the perforated steel plate opening constant, when the thickness of the steel ring gradually increases from 3 mm to 4 mm, 5 mm, 6 mm, 7 mm and finally, 8 mm, the thickness of the corresponding concrete tenon gradually decreases from 12 mm to 11 mm, 10 mm, 9 mm, 8 mm and finally, 7 mm, respectively, the yield load (*V_y_*) of the connector increases by 1.6%, 8.1%, 6.0%, 8.5% and 11.9%, accordingly, while the shear stiffness (*K_s_*) increases by 1.4%, 8.0%, 5.2%, 7.2% and 10.4%, respectively, with the yield load and shear stiffness gradually increasing as the thickness of the steel ring increases. From Figure 10b, it can be found that when the thickness of the steel ring gradually increases from 3 mm to 4 mm and 5 mm, the shear-bearing capacity (*V_u_*) of the connector increases by 2.7% and 6.8%, respectively, which indicates that the increase of the thickness of the steel ring can effectively improve the shear-bearing capacity of the connector. Meanwhile, when the thickness of the steel ring increases to 6 mm and 7 mm, respectively, the shear load capacity of the connector only increases by 1.3% and 1.1%, accordingly, and when the thickness of the steel ring increases to 8 mm, the shear load capacity of the connector is reduced by 2.4%, which is due to the fact that the thickness of the concrete tenon in the opening is too small to play a good shear load capacity. From Figure 10c to Figure 10h, as the thickness of the steel ring increases, the stress in the lower part of the steel ring and rubber ring, as well as the stress in the shear of the perforated rebar, the stress in the lower part of the concrete mortise and the stress in the upper part of the perforated steel plate opening all gradually reduce. This is because the increase in the thickness of the steel ring increases the shear area of the steel; therefore, the stress amplitude and deformation in the middle of the perforated rebar are reduced.

### 6.3. Influence of Rubber Ring Thickness

The load–slip curve of the PBL connector with steel–rubber tenon at different rubber thicknesses and the stress variation in each part of the specimen are shown in Figure 11.

It can be seen in Figure 11a that by keeping the diameter of the perforated steel plate opening constant, when the thickness of the rubber ring gradually increases from 3 mm to 4 mm, 5 mm, 6 mm, 7 mm and 8 mm, the thickness of the corresponding concrete tenon gradually decreases from 12 mm to 11 mm, 10 mm, 9 mm, 8 mm and 7 mm, respectively. The yield load (*V_y_*) of the connector is reduced by 1.8%, 2.4%, 3.4%, 3.5% and 3.5%, accordingly, and the shear stiffness (*K_s_*) is reduced by 11.1%, 15.4%, 27.3%, 33.4% and 38.7%, respectively, with the yield load and shear stiffness decreasing as the rubber ring thickness increases. From Figure 11b, it can be found that as the thickness of the rubber ring gradually increases from 3 mm to 4 mm, 5 mm, 6 mm, 7 mm and 8 mm, the shear load capacity (*V_u_*) of the connector decreases by 2.0%, 0.7%, 5.4%, 6.7% and 7.0%, respectively, which indicates that the increase in the thickness of the rubber ring causes a decrease in the shear load capacity of the connector under the condition that the diameter of the perforated steel plate opening remains the same. It indicates that the increase in the thickness of the rubber ring leads to a decrease in the thickness of the concrete tongue and groove, which in turn, reduces the shear load capacity of the connector and provides that the diameter of the perforated steel plate opening remains the same. From Figure 11c to Figure 11h, it can be seen that as the thickness of the rubber ring increases, the stress in the lower part of the steel ring and upper part of the rubber ring, as well as the stress at the shear of the perforated rebar, the stress at the upper part of the concrete tenon and the stress at the upper part of the perforated steel plate opening all gradually increase. This is because, without changing the perforated steel plate aperture, the thickness of the concrete tenon decreases as the thickness of the rubber ring increases, which increases the ductility of the internal tenon construction and reduces the stress concentration phenomenon, increasing the area of high-stress distribution between the steel ring and rubber ring as well as the stress amplitude.

### 6.4. Influence of Perforated Rebar Diameter

The load–slip curve of the PBL connector with steel–rubber tenon at different perforated rebar diameters and the stress variation in each part of the specimen are presented in Figure 12.

In Figure 12a, by keeping the perforated steel plate opening diameter constant, when the perforated rebar diameter gradually increases from 16 mm to 18 mm, 20 mm, 22 mm and 25 mm, the corresponding concrete tenon thickness gradually decreases from 12 mm to 11 mm, 10 mm, 9 mm and 7.5 mm, respectively. The yield load (*V_y_*) of the connector increases by 0.9%, 15.2%, 17.0% and 27.3%, accordingly, and the shear stiffness (*K_s_*) increases by 1.2%, 16.3%, 17.5% and 24.4%, respectively, i.e., the shear stiffness of the connector gradually increases with the increase of the perforated rebar diameter. In Figure 12b, as the perforated rebar diameter gradually increases from 16 mm to 18 mm, 20 mm, 22 mm and 25 mm, the shear load capacity (*V_u_*) of the connector increases by 1.7%, 10.4%, 6.1% and 11.7%, respectively, which indicates that by increasing the diameter of the perforated rebar, the shear load capacity of the connector can be effectively increased, though the concrete tenon thickness is reduced. From Figure 12c to Figure 12h, it can be seen that as the diameter of the perforated rebar increases, the stresses in the upper part of the steel ring, the upper part of the perforated rebar in shear and the lower part of the concrete tenon all decrease gradually; the stresses in the lower part of the steel ring, the upper part of the rubber ring, the upper part of the concrete tenon and the upper part of the perforated steel plate all increase gradually. The stresses in the lower part of the steel ring, the upper part of the rubber ring, the upper part of the concrete tenon and the upper part of the perforated steel plate openings all gradually increase. This is because when keeping the diameter of the perforated steel plate constant, as the diameter of the perforated rebar increases, the inner diameter of the steel ring and rubber ring increases, and the thickness of the concrete tenon decreases, which causes the concrete tenon to be damaged at a smaller load level. At the same time, the gradual increase in the diameter of the perforated rebar will result in a larger area of contact between the steel ring and the perforated rebar and the nearby concrete, reducing the bending deformation of the steel ring and the perforated rebar in shear, reducing the shear deformation of the steel ring and the perforated rebar, which makes them less susceptible to shear deformation.

### 6.5. Influence of Concrete Strength

The load–slip curve of the PBL connector with steel–rubber tenon at different strengths of concrete and the stress variation in each part of the specimen is proposed in Figure 13.

In Figure 13a, it can be seen that when the strength of the concrete gradually increases from 40 MPa to 50 MPa, 60 MPa, 70 MPa and 80 MPa, the yield load (*V_y_*) of the connector increases by 1.5%, 10.9%, 14.9% and 20.0%, respectively, and the shear stiffness (*K_s_*) increases by 1.5%, 0.8%, 4.5% and 9.1%, respectively, which is due to the fact that the yield load and shear stiffness of the connector gradually increase as the strength of concrete increases and its compressive capacity gradually increases as well. From Figure 13b, as the strength of concrete gradually increases from 40 MPa to 50 MPa, 60 MPa, 70 MPa and 80 MPa, the shear-bearing capacity (*V_u_*) of the connector increases by 13.0%, 19.3%, 27.9% and 34.8%, respectively, which indicates that the strength of concrete is increased, which can effectively increase the shear-bearing capacity of the connector. From Figure 13c to Figure 13h, it can be seen that as the strength of concrete increases, the stresses of the internal steel ring, rubber ring, perforated rebar shear, concrete mortise and upper area at the perforated steel plate opening all increase gradually. The stresses in the shear of the perforated rebar, the concrete tenon and the upper part of the perforated steel plate opening all increase gradually with the increasing strength of the concrete. This is due to the fact that the increase in the concrete tenon and the surrounding strength of concrete can synergize with the other tenon structures to exert a greater load-bearing capacity before reaching the critical state so that the connector obtains a superior shear-bearing performance. When the connector reaches the shear load capacity, the role of the steel ring, rubber ring and other tongue and groove structures are all better utilized at the same time, which makes the stress amplitude larger.

### 6.6. Influence of Perforated Steel Plate Strength

The load–slip curve of the PBL connector with steel–rubber tenon at different strengths of the perforated steel plate and the stress variation in each part of the specimen is seen in Figure 14.

From Figure 14a, it can be seen that when the strength of the perforated steel plate gradually increases from 235 MPa to 345 MPa, 390 MPa, 420 MPa and 460 MPa, the yield load (*V_y_*) of the connector increases by 9.0%, 8.5%, 9.9% and 10.1%, respectively, and the shear stiffness (*K_s_*) increases by 5.6%, 6.4%, 6.6% and 10.1%, respectively, due to the fact that as the strength of the perforated steel plate increases, the perforated steel plate can transmit larger loads with less deformation. From Figure 14b, it can be seen that as the strength of the perforated steel plate gradually increases from 235 MPa to 345 MPa, 390 MPa, 420 MPa and 460 MPa, the shear-bearing capacity (*V_u_*) of the connector does not change significantly, which is only −0.3%, −0.6%, 0.7% and 4.0%, respectively, which is due to the damage of the connector occurring in the form of shear damage of the perforated rebar; therefore, the increase in the strength of the perforated steel plate cannot better enhance the shear-bearing capacity of the connector. From Figure 14c to Figure 14h, as the strength of the perforated steel plate increases, the stresses on the steel ring, rubber ring and concrete tenon inside the specimen do not change significantly; the stress on the shear of the perforated rebar and the stress on the upper part of the perforated steel plate opening both gradually increase with the increasing strength of the perforated steel plate. This is because, as the strength of the perforated steel plate increases, the perforated steel plate is less likely to be deformed when subjected to shear loading; however, as the internal tenon construction remains unchanged and is damaged earlier than the perforated steel plate when subjected to shear, the stresses are on the perforated area. The stresses in the rebar shear and the upper part of the perforated steel plate opening increase gradually, while the stresses in the remaining parts do not change significantly.

## 7. Prediction of Shear Capacity

### 7.1. Previous Expressions

In this parametric study, to better test the shear resistance of the connector, having a clearance of 50 mm on the lower side of the perforated steel plate prevented inaccurate results due to the rigid base bearing pressure. Note that the friction between the perforated steel plate, steel beam and concrete is not considered. Several important studies and equations are reviewed in this section.

Based on the conditions and results set by the model, and considering the applicability of the current calculation expressions, through push-out tests of the PBL connector, and only considering the perforated steel plate and concrete, ignoring the perforated steel plate hole set rebars in the case, Leonhardt et al. investigated the damage criterion of the PBL connector and in Equation (1), the PBL connector shear-bearing capacity expression was proposed [44].
(1)$$V_{u}=1.4d_{p}^{2}f_{cu}$$
where $V_{u}$ is the shear-bearing capacity of a single perforated steel plate (N); $d_{p}$ is the opening diameter of the perforated steel plate (mm); and $f_{cu}$ is the compressive strength of the concrete cube (MPa).

Hosaka et al. compared the case of perforated steel plate with or without perforated rebar in the hole and studied the classification of the connector, and proposed Equation (2) to calculate the shear-bearing capacity without perforated rebar in the hole, which was considered in Equation (3) [45].
(2)$$V_{u}=3.38\sqrt{t_{p}/d_{p}}d_{p}^{2}f_{c}-39.0\times 10^{3}$$
(3)$$V_{u}=1.45\left[\left(d_{p}^{2}-d_{r}^{2}\right)f_{c}+d_{r}^{2}f_{ru}\right]-26.1\times 10^{3}$$
where $V_{u}$ is the shear load capacity of a single perforated steel plate (N); $t_{p}$ is the thickness of the perforated steel plate (mm); $d_{p}$ is the diameter of the opening in the perforated steel plate (mm); $f_{c}$ is the compressive strength of the concrete (MPa); $d_{r}$ is the diameter of the perforated rebar (mm); and $f_{ru}$ is the ultimate tensile strength of the perforated rebar (MPa). 

Zheng et al. proposed a slotted perforated steel plate connector, which was parametrically analyzed by push-out tests combined with numerical simulations, and Equation (4) was proposed to calculate its shear-bearing capacity [37].
(4)$$V_{u}=\gamma_{n}\gamma_{e}\left[0.42\left(d_{p}^{2}-d_{r}^{2}\right)f_{c}+1.15d_{r}^{2}f_{ry}+0.45d_{p}t_{p}f_{sy}\right]$$
where $V_{u}$ is the shear load capacity of a single perforated steel plate (N); $t_{p}$ is the thickness of the perforated steel plate (mm); $d_{p}$ is the diameter of the opening in the perforated steel plate (mm); $f_{c}$ is the compressive strength of the concrete (MPa); $d_{r}$ is the diameter of the perforated rebar (mm); $f_{ry}$ is the yield strength of the perforated rebar (MPa); $f_{sy}$ is the yield strength of the steel (MPa); $\gamma_{n}$ is the coefficient considering the effect of the number of holes in the steel plate, $\gamma_{n}=n_{p}^{-0.22}$, $n_{p}$ is the number of holes; $\gamma_{e}$ is the coefficient considering the effect of the spacing of the holes in the steel plate, $\gamma_{e}=1+0.002\cdot \left(e_{p}-200\right)\leq 1$, in which $e_{p}$ is the hole spacing.

### 7.2. Proposed Expression

Currently, the proposed PBL connector with steel–rubber tenon as a new type of shear-resistant connector has not yet been studied theoretically. This section, based on the previous numerical simulation, refers to the existing research results, and in order to draw on the introduction process of the above equation, it can be concluded that the connector is mainly damaged due to the shear load of the perforated rebar and internal tenon structure that mainly includes steel ring, rubber ring and concrete tenon. The shear load capacity of the connector is related to the diameter of the perforated rebar, the thickness of the steel ring, concrete tenon and rubber ring, based on the above analysis and the results of the parametric study.

Therefore, this section proposes an alternative expression for this new PBL connector with steel–rubber tenon. Equations (5) and (6) combine the above four influencing factors to predict the shear load capacity of this type of connector.
(5)$$V_{u}=D_{1}A_{s}f_{sy}+D_{2}A_{c}f_{c}+D_{3}A_{s}^{\prime }f_{sy}^{\prime }$$
(6)$$V_{u}=D_{1}d_{r}^{2}f_{sy}+D_{2}\left[d_{p}^{2}-{\left(d_{r}+2t_{s}+2t_{r}\right)}^{2}\right]f_{c}+D_{3}\left(d_{r}+t_{s}\right)t_{s}f_{sy}^{\prime }$$
where $V_{u}$ is the shear load capacity of a single perforated steel plate (N); $A_{s}$ is the cross-sectional area of the perforated rebar (mm^2^); $A_{c}$ is the cross-sectional area of the concrete tenon (mm^2^); $A_{s}^{\prime }$ is the cross-sectional area of the steel ring (mm^2^); $d_{p}$ is the opening diameter of the perforated steel plate (mm); $d_{r}$ is the diameter of the perforated rebar (mm); $t_{s}$ is the thickness of the steel ring (mm); $t_{r}$ is the thickness of the rubber ring (mm); $f_{sy}$ is the yield strength of the perforated rebar (MPa); $f_{c}$ is the compressive strength of the concrete (MPa); $f_{sy}^{\prime }$ is the yield strength of the steel ring (MPa) and $D_{1}$, $D_{2}$ and $D_{3}$ are the three fitting parameters of the equation, respectively.

Combined with the results of the parametric analysis presented in this paper for Equation (6) for the nonlinear regression analysis yields, when $D_{1}$ = 1.230, $D_{2}$ = 2.598, $D_{3}$ = 3.085 and *R*^2^ = 0.9983, indicating that the fit of the regression line, at this time, is high and can be used for this alternative equation to calculate the shear-bearing capacity of the connector, where the specific equation is given by Equation (7).
(7)$$V_{u}=1.23d_{r}^{2}f_{sy}+2.598\left[d_{p}^{2}-{\left(d_{r}+2t_{s}+2t_{r}\right)}^{2}\right]f_{c}+3.085\left(d_{r}+t_{s}\right)t_{s}f_{sy}^{\prime }$$

The shear load capacity, predicted by Equation (7) as well as the results of the finite element simulation, are compared and verified in Figure 15.

As shown in Figure 15, the simulation and the predicted results are in good agreement, which can provide a reference for the subsequent research of push-out tests and the promotion of the test pieces.

## 8. Conclusions

A new PBL connector with steel–rubber tenon is proposed in this paper, which aims to increase the connector’s deflection while ensuring the shear resistance of the connector.

To investigate its shear resistance property, first, the design of the connector was carried out. Second, the finite simulation method was validated with [41], and the modeling of different types of PBL connectors was carried out based on the validated finite element method, while the superiority of the force mechanism and mechanical properties of the PBL connector with steel–rubber tenon are discussed in detail through a comparative analysis. Finally, by parametrically analyzing the thickness of the steel ring and rubber ring, the diameter of the perforated rebar and the strength of the concrete and perforated steel plate of the connector, a new method for calculating the shear load capacity of PBL connectors with steel–rubber tenon is proposed. The following conclusions can be drawn:

(1) By comparing the mechanical properties of the PBL connector with a steel ring tenon and conventional PBL connector, it was found that the shear stiffness of the connector could increase by 9.8% and the shear load capacity increased by 9.2% when only the 5 mm thick steel ring is considered, which indicates that the steel ring enables the shear resistance of the connector to increase by increasing the cross-sectional shear area through only a small amount of steel. 

(2) By comparing the mechanical properties of the PBL connector with a steel ring tenon and the PBL connector with a steel–rubber tenon, it was found that the yield slip of the PBL connector with steel–rubber tenon is 175.1% of the PBL connector with steel ring tenon, which indicates that the setting of the rubber ring can significantly improve the ductility of the specimens.

(3) By comparing the mechanical properties of the PBL connector with steel–rubber tenon and conventional PBL connector, it was found that the shear stiffness of the connector was reduced by 37.3%, and the shear load capacity increased by 13.7% by considering the addition of a 5 mm thick steel ring and a 5 mm thick rubber ring, and the deformation capacity and shear force distribution of the connector were optimized due to the small stiffness of natural rubber setting.

(4) The load–slip curves of PBL connectors with different tenon structures are divided into three stages: in the first stage, the slope of the curve is large, which shows the linear elasticity and high stiffness behavior without a large slip; in the second stage, the curve trend is arc-shaped, which shows a nonlinear and gradually decreasing stiffness behavior, where the load increases slowly with the slip, and at the point of the extreme value of the curve, the load starts to show a slow; in the third stage, the load slowly increases with the slip until the specimen reaches damage.

(5) Based on the finite element analysis results of the PBL connector with steel–rubber tenon, the effects of the thickness of the steel ring and rubber ring, the diameter of perforated rebar, and the strength of concrete and perforated steel plate are analyzed. The increase in rebar diameter, steel ring thickness and concrete strength will improve the connector’s shear stiffness and shear capacity. The increase in the thickness of the rubber ring can cause a decrease in the shear stiffness and the shear load-carrying capacity of the connector; however, the change in the strength of the perforated steel plate is negligible.

(6) By analyzing the existing shear load capacity calculation equation, referring to its derivation process, and combining the finite element parametric analysis results and the damage mechanism of the connector, the calculation equation applicable to the presented novel PBL connector with steel–rubber tenon is built to predict the shear load capacity of the connector. The calculated results are in good agreement with the finite element simulation results and can be referred to for further research and real application.

## Figures and Tables

**Figure 1 materials-16-02291-f001:**
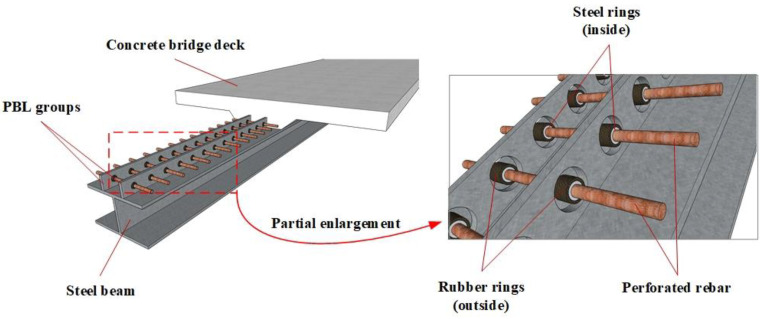
Application of the PBL connector with steel–rubber tenon.

**Figure 2 materials-16-02291-f002:**
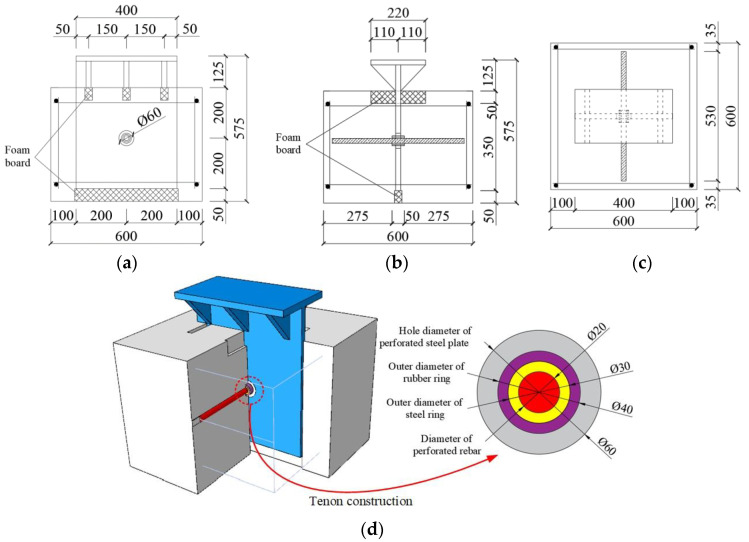
Specimen configuration of PBL connector with steel–rubber tenon/mm: (**a**) elevation view, (**b**) side view, (**c**) top view, (**d**) three-dimensional view.

**Figure 3 materials-16-02291-f003:**
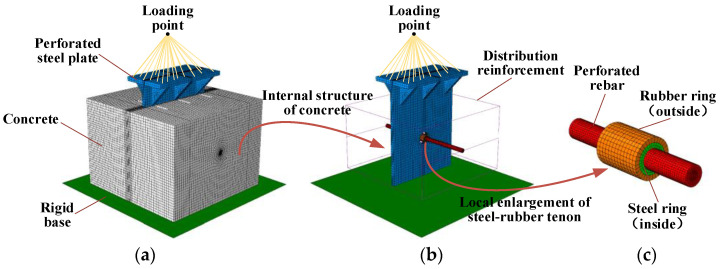
Finite element model and cell division: (**a**) overall model, (**b**) concrete internal structure, (**c**) partial construction with steel–rubber tenon.

**Figure 4 materials-16-02291-f004:**
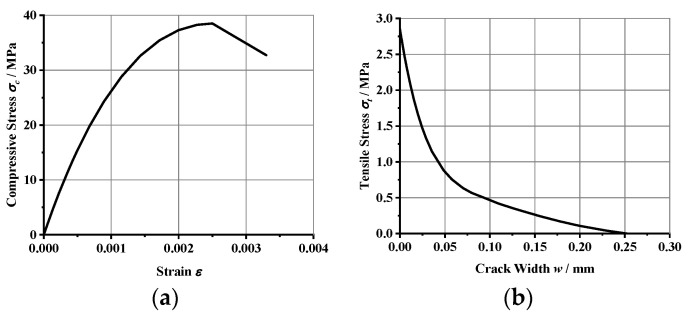
Concrete constitution: (**a**) concrete in compression, (**b**) concrete in tension.

**Figure 5 materials-16-02291-f005:**
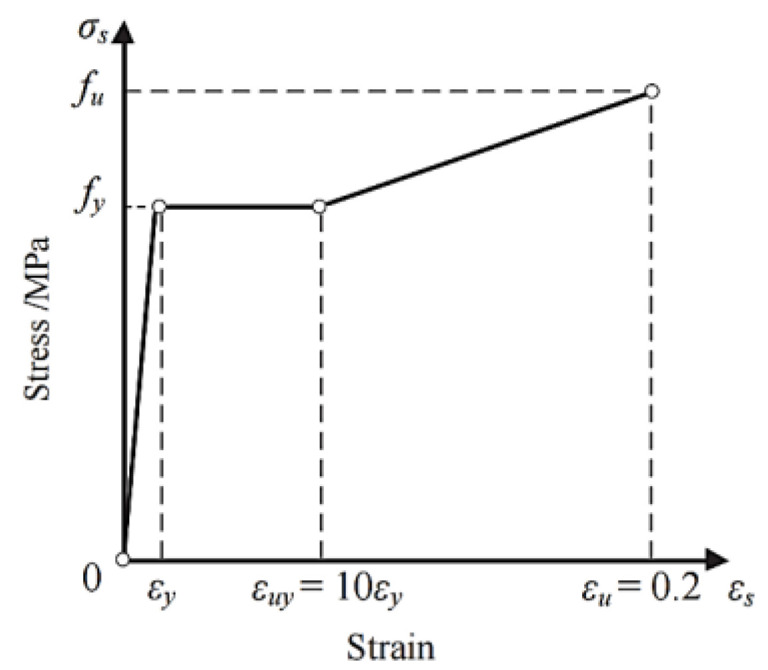
Steel constitution.

**Figure 6 materials-16-02291-f006:**
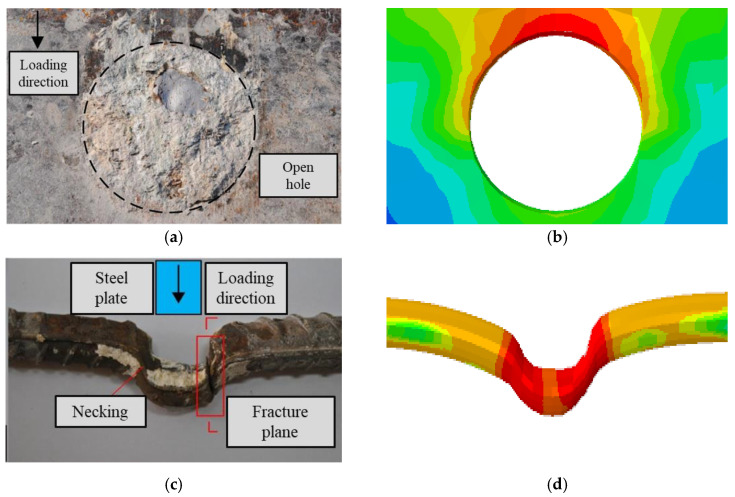
Comparison of damage patterns: (**a**) perforated steel plate (test), (**b**) perforated steel plate (FEA), (**c**) perforated rebar (test), (**d**) perforated rebar (FEA).

**Figure 7 materials-16-02291-f007:**
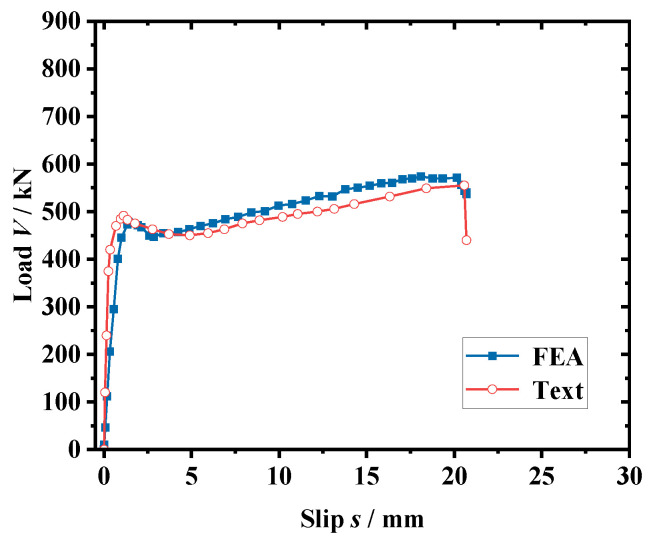
Comparison of load–slip curve.

**Figure 8 materials-16-02291-f008:**
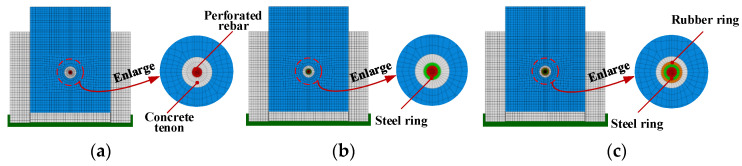
Cutaway of three different types of shear connector models: (**a**) concrete tenon, (**b**) steel ring tenon, (**c**) steel–rubber tenon.

**Figure 9 materials-16-02291-f009:**
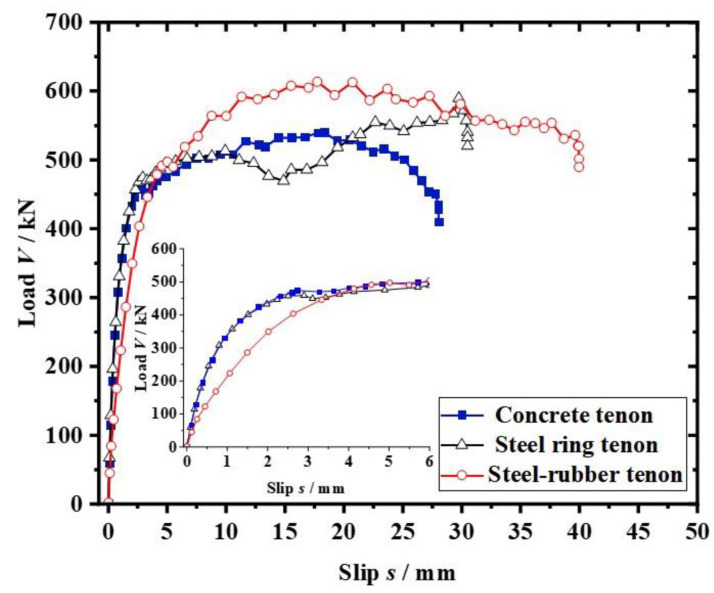
Three different types of load–slip curves.

**Figure 10 materials-16-02291-f010:**
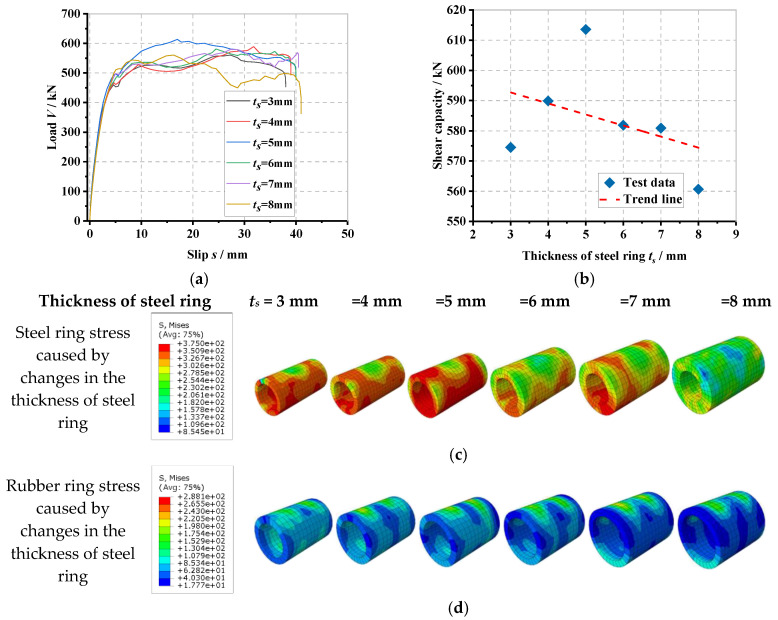
Influence of steel ring thickness: (**a**) overall load–slip curves, (**b**) impact analysis, (**c**) steel ring stress, (**d**) rubber ring stress, (**e**) concrete tenon stress, (**f**) perforated rebar stress, (**g**) perforated steel plate stress, (**h**) overall stress in the specimen.

**Figure 11 materials-16-02291-f011:**
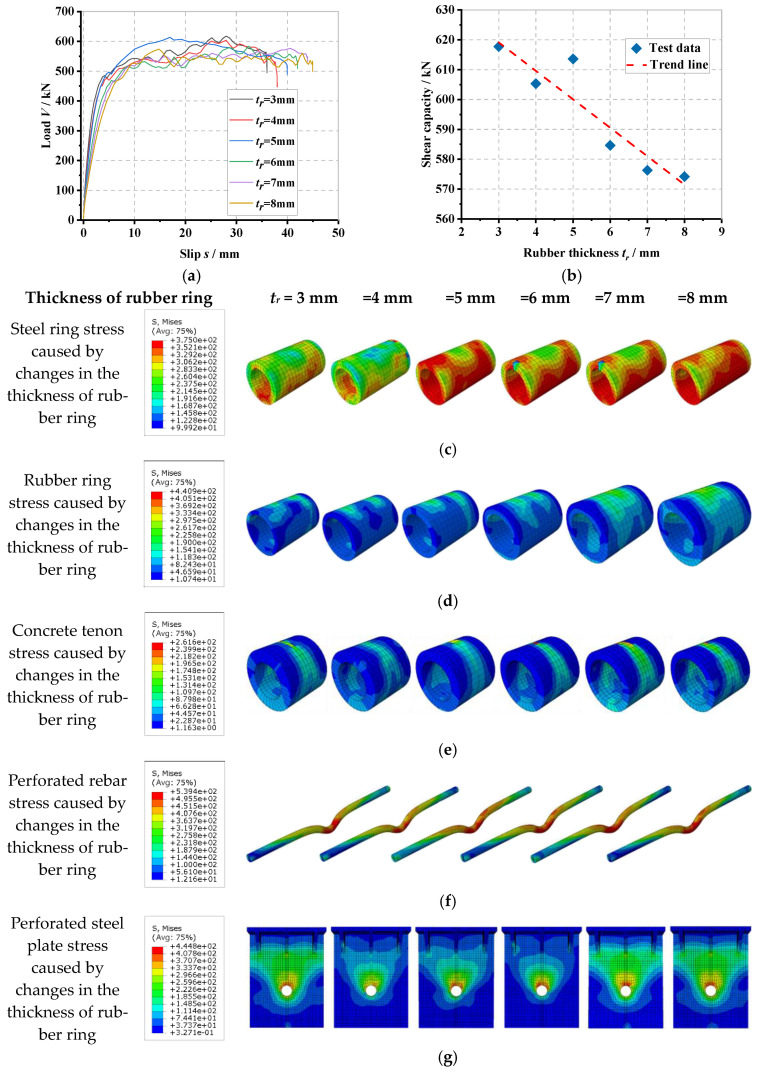
Influence of rubber ring thickness: (**a**) overall load–slip curves, (**b**) impact analysis, (**c**) steel ring stress, (**d**) rubber ring stress, (**e**) concrete tenon stress, (**f**) perforated rebar stress, (**g**) perforated steel plate stress, (**h**) overall stress in the specimen.

**Figure 12 materials-16-02291-f012:**
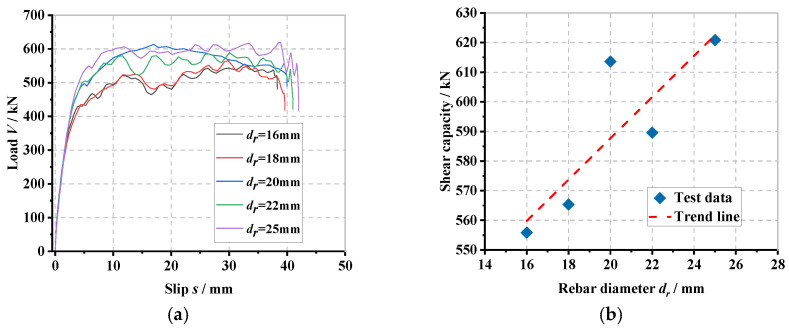
Influence of perforated rebar diameter: (**a**) overall load–slip curves, (**b**) impact analysis, (**c**) steel ring stress, (**d**) rubber ring stress, (**e**) concrete tenon stress, (**f**) perforated rebar stress, (**g**) perforated steel plate stress, (**h**) overall stress in the specimen.

**Figure 13 materials-16-02291-f013:**
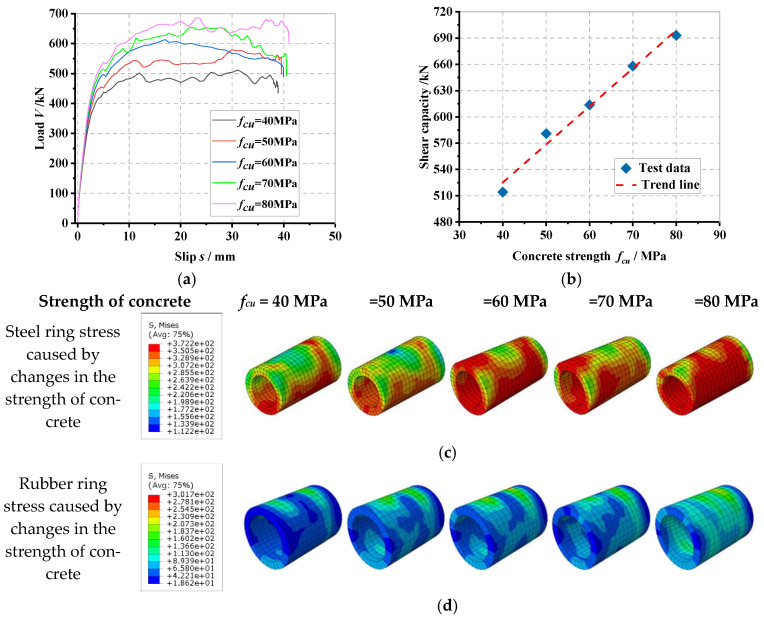
Influence of concrete strength: (**a**) overall load–slip curves, (**b**) impact analysis, (**c**) steel ring stress, (**d**) rubber ring stress, (**e**) concrete tenon stress, (**f**) perforated rebar stress, (**g**) perforated steel plate stress, (**h**) overall stress in the specimen.

**Figure 14 materials-16-02291-f014:**
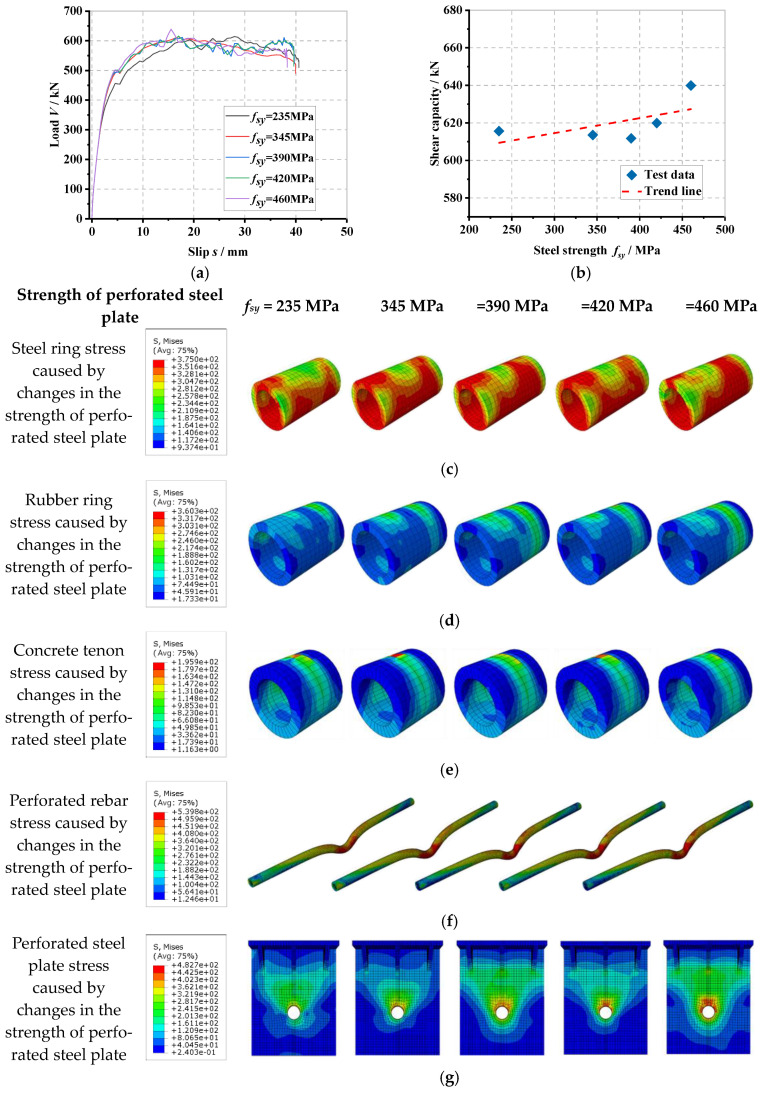
Influence of perforated steel plate strength: (**a**) overall load–slip curves, (**b**) impact analysis, (**c**) steel ring stress, (**d**) rubber ring stress, (**e**) concrete tenon stress, (**f**) perforated rebar stress, (**g**) perforated steel plate stress, (**h**) overall stress in the specimen.

**Figure 15 materials-16-02291-f015:**
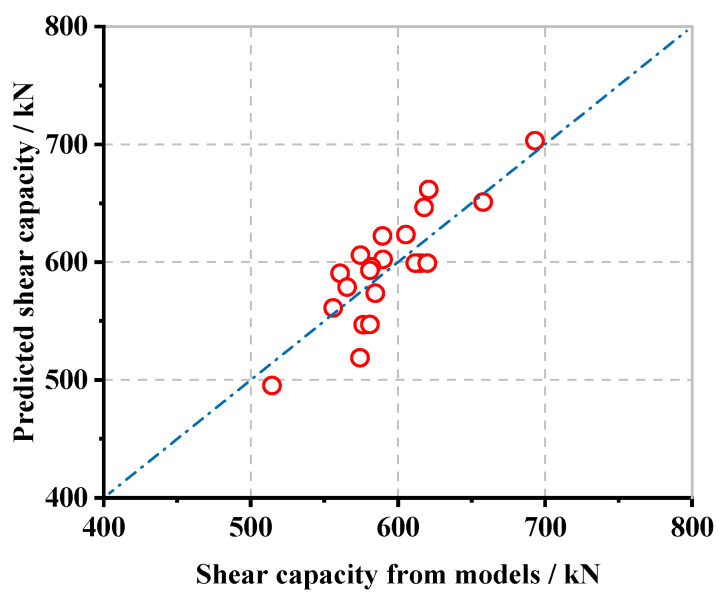
Comparative verification of shear-bearing capacity equations.

**Table 1 materials-16-02291-t001:** Table of steel material properties.

Material	Modulus of Elasticity *E*_s_/Gpa	Yield Strength ƒ_y_/MPa	Ultimate Strength ƒ_u_/MPa	Poisson’s Ratio *μ*
HRB400 rebars	210	400	540	0.3
Q345 steel plate		345	470	
Q235 steel ring		235	375	

**Table 2 materials-16-02291-t002:** Table of intrinsic rubber parameters.

Material	*C*_10_/MPa	*C*_20_/MPa	*D*_1_/MPa^−1^	*D*_2_/MPa^−1^
Natural rubber	29.4	0.72	0.0017	0

**Table 3 materials-16-02291-t003:** Summary of parametric study results.

Model	*t_s_* (mm)	*t_r_* (mm)	*d_r_* (mm)	*ƒ_cu_* (MPa)	*ƒ_sy_* (MPa)	*K_s_* (kN/mm)	*V_y_* (kN)	*V_u_* (kN)
TS-3	3	5	20	60	345	91.76	459.83	574.53
TS-4	4	5	20	60	345	93.02	467.01	589.91
TS-5	5	5	20	60	345	99.13	497.17	613.59
TS-6	6	5	20	60	345	96.53	487.59	581.85
TS-7	7	5	20	60	345	98.39	498.82	580.90
TS-8	8	5	20	60	345	101.29	514.65	560.66
TR-3	5	3	20	60	345	117.12	485.40	617.71
TR-4	5	4	20	60	345	104.17	476.69	605.33
TR-5	5	5	20	60	345	99.13	497.17	613.60
TR-6	5	6	20	60	345	85.15	468.92	584.64
TR-7	5	7	20	60	345	78.04	468.54	576.29
TR-8	5	8	20	60	345	71.81	468.42	574.20
DR-16	5	5	16	60	345	85.26	431.53	555.86
DR-18	5	5	18	60	345	86.32	435.48	565.37
DR-20	5	5	20	60	345	99.13	497.17	613.60
DR-22	5	5	22	60	345	100.22	504.77	589.57
DR-25	5	5	25	60	345	106.04	549.54	620.85
CU-40	5	5	20	40	345	98.31	448.46	514.29
CU-50	5	5	20	50	345	99.74	455.01	580.98
CU-60	5	5	20	60	345	99.13	497.17	613.60
CU-70	5	5	20	70	345	102.78	515.49	657.98
CU-80	5	5	20	80	345	107.27	538.01	693.05
SY-235	5	5	20	60	235	93.84	456.22	615.60
SY-345	5	5	20	60	345	99.13	497.17	613.60
SY-390	5	5	20	60	390	99.87	494.87	611.76
SY-420	5	5	20	60	420	99.99	501.49	619.90
SY-460	5	5	20	60	460	103.30	502.19	639.94

Note: TS is the thickness of steel ring, TR is the thickness of rubber ring, DR is the diameter of perforated rebar, CU is the strength of concrete, and SY is the strength of the steel plate in the table.

## Data Availability

All data generated or analyzed during this study are included in this article. All data included in this study are available upon request by contact with the corresponding author.

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
