# Peer review of "Study on Shear Resistance Property of a New PBL Connector with Steel–Rubber Tenon"

_materials, 2023, doi:10.3390/ma16062291_

Round 1

Reviewer 1 Report

The study titled "Study on shear resistance property of a new PBL connector with steel-rubber tenon" aims to improve the shear load capacity and ductility of PBL connectors by proposing a new connector with a steel-rubber tenon. The study uses finite element modeling and mechanical property comparison to show that the proposed connector can effectively improve the stiffness and shear load capacity of the PBL connector while also reducing the shear stiffness and improving the deformation capacity of the specimen.

The research also analyzes the effects of various parameters on the performance of the new connector and presents a calculation equation to predict the shear load capacity of the connector. Overall, the study presents a comprehensive analysis of the proposed connector's properties and provides useful insights for improving the design of PBL connectors.

The paper is interesting and could be published in the Journal of materials in the reviewer's view if the following comments and other reviewers, comments to be fulfilled.

When using section headings, it is important to adhere to proper formatting conventions. Specifically, a section heading should not be labeled as "0. Introduction" as this implies that there is no preceding content or background information. It is recommended to use a numbering system that begins with "1. Introduction" to accurately reflect the sequential order of the content.

What are we hoping to accomplish with this study?

The PBL connector with the steel-rubber tenon is defined as the following:

What makes the new PBL connector with steel-rubber tenon stand out from existing shear connectors on the market?

The background information in the introduction is insufficient and does not provide a comprehensive understanding of the study's context and potential applications. The literature review is inadequate and requires more focus on the research work, including a thorough explanation of the entire process and its past, present, and future scope. Although the authors have cited some relevant references to outline the research problem, the lack of recent references may indicate a lack of a comprehensive literature review. It would be beneficial to refer to below recent and up-to-date research papers related to the topic, particularly in recent years, in order to strengthen the study's foundation:

-Application of support vector machine with firefly algorithm for investigation of the factors affecting the shear strength of angle shear connectors

- Application of ANFIS technique on performance of C and L shaped angle shear connectors

- Investigation of pipe shear connectors using push out test

- Strengthening of bolted shear joints in industrialized ferrocement construction

The approved finite element approach refers to what exactly?

How did the new PBL connector model specifications come to be developed?

What is the purpose of the push-out test and the beam test in testing the shear resistance of the new PBL connector?

How can the test results from the push-out test and the beam test be compared and interpreted for the new PBL connector?

How are the various parts of the model represented in the simulation, and what are the benefits of using C3D8R and T3D2 cells?

How is the distribution reinforcement used to restrain the concrete and form a reinforced concrete structure, and what type of cells are used to model its effects?

How are the concrete compressive stress-strain curve and tensile stress-crack width relationship curves used in the finite element analysis method?

What is the key construction of the new connector, and what is the aim of the authors in testing its force performance?

How are the load-slip curves for the new PBL connector with steel-rubber tenon, the PBL connector with steel ring tenon, and the conventional PBL connector simulated and presented in Figure 9?

How does the yield slip Sy of the load-slip curves compare between the three connectors under ultimate load?

How does the shear stiffness Ks of the load-slip curves compare between the three connectors under ultimate load?

How do the load-slip curves and shear stiffness values obtained from the finite element analysis method compare to the results obtained from the push-out test and beam test in the previous section of the text?

What materials are used in making the test piece, the steel ring, the rubber ring, the perforated steel plate, and the concrete slab?

What are the dimensions of the steel ring, the rubber ring, and the perforated steel plate, and what is the size of the concrete slab?

Is the concrete slab equipped with any reinforcement mesh, and if so, what is its purpose?

How is the layer of reinforcement mesh in the concrete slab expected to affect the test results?

How do the dimensions of the steel ring, rubber ring, and perforated steel plate affect the force performance of the new connector?

How is the C60 concrete used in making the concrete slab different from other types of concrete?

How does the thickness of the steel ring and the perforated steel plate impact the shear resistance of the new connector?

Why is the push-out test considered to give a lower limit of the shear load capacity of the connector?

Reviewer 2 Report

The authors should put a tremendous effort to do the required comments. My comments are listed as follows:
- The abstract has a poor presentation; it should be re-written to include the studied parameters, the methodology, and the important findings.
- The introduction is too short, must be much improved and recent researches should be added. The introduction needs more attention than the current situation to show the gap between what has been studied and the importance of the current research.
- There are several studies reporting similar data but authors offer no comparison with these results (comparison of results with past mentioned above studies are required).
- Meaningful conclusions are needed as conclusions are general and is common sense. The conclusion must be reconstructed & simplified into 6-7 important points with supporting results.

- How can the results of this paper help to improve the specification?
- The English grammar needs careful attention and correction.

Round 2

Reviewer 2 Report

The manuscript has been adequately revised.